# Deep Learning for Compute in Memory

## ABSTRACT

Compute in Memory (CIM) accelerators for neural networks promise large efficiency gains, allowing for deep learning applications on extremely resource-constrained devices. Compared to classical digital processors, computations on CIM accelerators are subject to a variety of noise sources such as process variations, thermal effects, quantization, and more. In this work, we show how fundamental hardware design choices influence the predictive performance of neural networks and how training these models to be hardware-aware can make them more robust for CIM deployment. Through various experiments, we make the trade-offs between energy efficiency and model capacity explicit and showcase the benefits of taking a systems view on CIM accelerator and neural network training co-design.

**ACM Reference Format:**
Anonymous Author(s). 2020. Deep Learning for Compute in Memory. In *Online '21: tinyML Research Symposium, March 22–26, 2020, Online.* ACM, New York, NY, USA, 8 pages. https://doi.org/10.1145/nnnnnnn.nnnnnnn

## 1 INTRODUCTION

Neural network applications are moving more and more away from data-centers into the end-users' devices such as smartphones, drones, wearables, and other IoT devices. This so-called edge computing offers distinct advantages compared to centralized computing in the cloud such as data-privacy, low-latency, and no dependence on internet connectivity at all. On the other hand, edge devices can be extremely resource-constrained. Applications for edge computing range from hearing-aids, always-on applications like key-word spotting to battery-driven security cameras. For all of these applications, we aim to maximize "AI Performance per Watt", where AI performance is a task-dependent metric such as accuracy or false detection rate. In trying to optimize AI performance per Watt, efforts are usually split across researchers or engineers working on optimizing hardware (operations per Watt) and the machine learning community that is trying to optimize models for a given hardware (AI performance per operations). In this paper, we propose taking a unified view on the co-optimization of hardware and neural networks based on a differentiable hardware simulator for highly efficient Compute in Memory (CIM) architectures. CIM computations require orders of magnitudes less energy and are substantially faster in applications where the classical von Neumann architecture hits its limits due to excessive memory transfers [22].

Neural networks are a prime example of such applications. Making predictions with a neural network requires the loading of large weight matrices into memory, as well as the transportation of intermediate computations. Even on specialized digital hardware, such as GPUs or TPUs, the working memory of their parallel processors needs to be constantly updated from off-chip memory, incurring high energy costs and latencies. The flip side of CIM computations is that they are fundamentally different from standard digital computations. Several sources of noise, such as thermal fluctuations and process variations might cause an analog processor to compute a different function between subsequent executions and between two seemingly identical chips. Whereas in digital processors, these noisy physical processes are abstracted away into digital units of zero and one and paired with digital error correction, CIM computations are inherently noisy. As such, when mentioning a "CIM chip" in this work, we refer to a mixed-signal integrated circuit.

Neural networks, on the other hand, have proven to be quite tolerant of or even benefit from certain types of noise injections. In dropout [16], for example, injecting Bernoulli or Gaussian noise into the network's activations serves to avoid overfitting. Together, the memory intensive nature and noise resilience of neural networks promise large gains in efficiency when the model is executed on CIM hardware, opening up new possibilities for deploying large neural networks on resource-constrained devices.

In this work, we show that accessing the advantages of CIM computation for neural networks requires noise-aware training and careful consideration of several aspects of hardware and neural network architecture co-design. We provide an overview of how design choices influence predictive performance in the context of minimizing energy expenditure. Empirical results validate our argumentation and provide intuition about the design space on CIFAR-10 [10] and the google speech commands dataset [20].

## 2 HARDWARE

*CIM array.* When designing hardware for neural network acceleration, the measure of success is operations per Watt. Compute in Memory architectures for neural networks amortize expensive memory access by computing individual input-weight multiplications in memory. Instead of moving inputs, weights, and instructions to a digital compute unit in sequence, a matrix-vector multiplication in-memory is performed in one cycle without moving weights. In typical CIM accelerators, each element of a weight matrix $w_{i,j}$ is placed on the intersection between a word-line, which carries the input activation information, and the bit-line, across which the result of the multiplication with the input $x_i$ and that particular column of the weight matrix is read out. See Figure 1 for a visualization of how convolutions are mapped to a series of matrix-vector multiplications within a CIM array. Since all columns are computed and read-out in parallel, a whole matrix-vector multiplication is performed in one cycle.

Different designs for realizing this approach in hardware have been proposed. Major dimensions that have been explored include

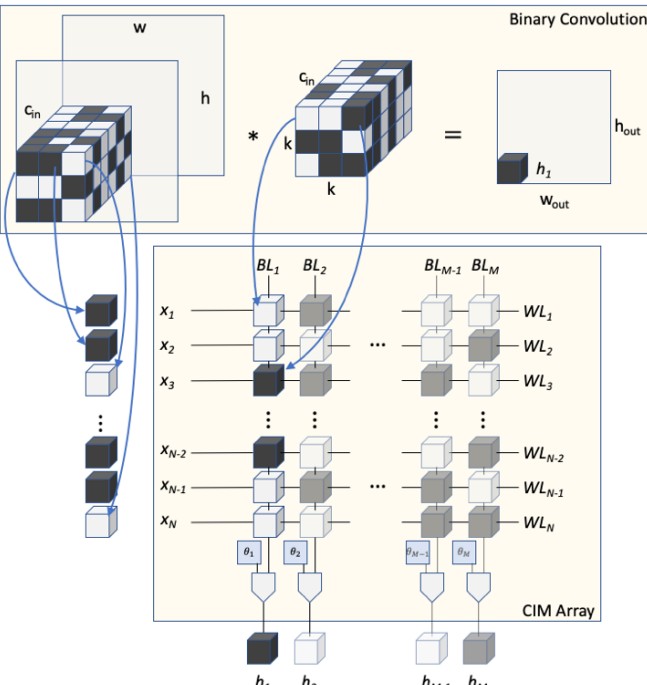

**Figure 1: Above: One convolutional kernel applied to an input patch. This results in one scalar entry in one of $M$ output feature maps. Below: CIM array crossbar design. The flattened kernel corresponds to one column of the array, the flattened input patch is presented on the word-lines.**

the choice of analog compute domain (Charge, Current, Voltage based) as well as memory technologies (SRAM, MRAM, RRAM). In this work, we consider the combination of SRAM and charge domain computation as a consequence of general consideration of technology maturity and sensitivity to process-voltage-temperature variations. Furthermore we restrict the individual bit-cell operation to binary input activations $x$ and weights $w$: $x, w \in \{0, 1\}$. We show in section 3.3 how binary matrix-vector multiplications can be used to enable multi-bit computations.

During a matrix-vector multiplication, the word-lines are activated, causing the capacitors in the bit-cells to charge dependent on the value of the input and stored weight according to the XNOR operation. Next, all capacitors in a column are shorted together, modifying the voltage on the corresponding bit line: For cells whose XNOR operation evaluates to 0, the bit-line voltage remains untouched, whereas for 1, the voltage increases by redistributing the accumulated charges over those XNOR cells. For a CIM array with $N$ rows, the pop-count lies between 0 (all cells 0) and $N$ (all cells 1). In Section 3.2 we make explicit how we map matrix-vector multiplications within a neural network to a CIM array with these capabilities. The SRAM-based charge domain approach has shown to exhibit a high-fidelity linear relationship between voltage and pop-count [1], however, due to cap mismatch and thermal effects, this relationship is subject to noise. Furthermore, the maximal voltage increase across a bit-line is independent of the number of word-lines within the array. As the chip size increases, a difference in one pop-count corresponds to a smaller absolute difference of voltage as measured at the bit-line. Computation on a CIM array is therefore subject to a

signal-to-noise ratio influenced by the maximum voltage difference, array size, and noise magnitudes.

*ADC.* After a linear operation, a neural network layer's output is optionally transformed via operations such as batch normalization [6], pooling, or passed through a non-linearity. Since we consider binarization of the input to the next layer, applying batch normalization to the result of the previous layer is equivalent to a simple comparison of the pop count with a specific threshold [14, 18].

The combination of applying a convolution followed by batch normalization and subsequent binarization for the next layer is therefore equivalent to comparing the bit-line voltage to a threshold voltage. This binary operation is performed using a comparator hardware element. Other digital operations, such as fusing computations across several CIM arrays, average pooling, or performing argmax across columns at the output layer require access to the actual pop count. This digitization operation is performed using an analog-to-digital converter (ADC) with $b$ bit resolution. An ADC digitizes the bit-line voltage to one of $2^b$ values in $[0, N]$. Depending on the number of rows $N$ in the array, $b$ bits might not be enough to distinguish individual pop count values, resulting in quantization noise. The energy costs of ADC evaluations increase exponentially with the required bit-width $b$ [11]. The optimal bit-width $b$ will therefore be a trade-off between quantization noise tolerance and energy constraints. We show how quantization noise as a consequence of applying ADC with limited bit-widths influences neural network performances in Section 5.4.

*CIM Chip.* A fundamental step in designing a CIM accelerator is to map a given neural network to its components. More specifically, one has to choose the number of individual CIM arrays to place into the accelerator and how to map layers of a neural network to the individual arrays. We consider three scenarios: CIM arrays with the same size as a layer's kernel volume, CIM arrays with a predetermined number of rows (**non-shared**) and a CIM chip with one shared CIM array of fixed size (**shared**). Whereas in the first two cases, we assume to have as many arrays available as required for a given network architecture, the shared scenario requires unloading and loading weights in order to compute the output of the network. For a general-purpose CIM chip, the size of an arbitrary neural network will possibly exceed any fixed number of available CIM arrays. The shared and non-shared approaches therefore represent two extremes, in between which there is a chip configuration with several CIM arrays, of which some are shared. The first approach will serve as a baseline to study the other two.

We show experimentally that the number and size of CIM arrays amongst other factors significantly influence the predictive performance of a neural network. Product design decisions will have to weigh these predictive performance characteristics with the power profile of the resulting chip to find an acceptable balance.

*Not considered.* We explicitly do not take into account analog implementations of neural network elements other than matrix-vector multiplication and batch normalization in the binary case since those are the expensive operations in traditional neural network design. If a neural network contains elements other than these, their analog hardware implementations will have to be modeled or assumed to be performed in the digital domain.

# 3 METHOD

## 3.1 Hardware Simulation

A generic pre-trained neural network experiences a significant drop in predictive performance when naively mapped to a CIM chip (Section 5.1). Even if the pre-trained model has been trained to perform well when quantized to binary values, the differences between an all-digital computation and computations within a CIM chip cause the model to drop in performance (see Tables 1 and 2). To avoid a drop in performance when deploying a neural network, we aim to train models that are robust to the CIM specific differences to the digital domain. To expose the training procedure to these CIM noises and to understand the impact of hardware design choices on neural network performance, we develop a CIM simulator in three steps:

(1) Low-level SPICE simulations of a CIM array.
(2) Abstraction of the low-level noise model into a high-level differentiable CIM array simulator.
(3) Integration of the array simulator into a CIM chip simulator.

*SPICE simulations.* The SPICE simulations include a CIM array of $N$ word lines and a single bit line. The cell weights are randomly initialized to 0 or 1. Subsequently, all rows are activated in sequence by switching the corresponding word-line such that XNOR evaluates to 1. For each in this way activated word-line, the bit-line voltage corresponding to a pop-count from 0 to $N$ is being read out. After this bit-line voltage vs. pop-count characterization is done at a typical case, Monte Carlo simulations generate bit-line voltage variations at each individual pop-count in $[0, N]$ according to the hardware noise model.

*CIM array simulation.* Given the insights from these low-level simulations, we can characterize the CIM array noises acting on the clean pop-count computation $a$ for a particular column into three zero-mean normally distributed sources $\alpha, \beta$ and $\gamma$ such that

$$\tilde{a} = \alpha a + \beta + \gamma. \tag{1}$$

These noise sources originally influence computation in the voltage domain. We therefore translate a standard deviation $\sigma_\epsilon'$ from the voltage domain into the pop count domain through Equation (2). $\Delta v$ corresponds to the maximum voltage difference across a bit-line. $N$ corresponds to the number of rows in the CIM array.

$$\sigma_\epsilon = \frac{\sigma_{\epsilon'} \cdot N}{\Delta v} \qquad \left[ \frac{V \cdot Pcnt}{V} \right] \tag{2}$$

These noise sources are distinguished depending on their origin and how they affect $a$. Firstly, CIM computations are stochastic across the distribution of all CIM arrays produced. Stochasticity enters at the point of manufacturing in the form of process variations. For a given instantiation of a CIM array, a sample from these noise sources is drawn and influences the array's subsequent computations in a deterministic manner. Specifically, $\alpha$ covers charge cap variations and affects the pop-count computation as data-dependent multiplicative noise $\mathcal{N}(0, \sigma_\alpha(a))$. $\beta$ subsumes additive noises such as offset variations. Secondly, $\gamma$ subsumes noise sources such as thermal fluctuations that are applied to every read-out of a CIM array's column. The magnitude of these noise sources is determined by the entirety of hardware design and assumptions modeled in

step 1. Here we only consider noise sources of these types, however future iterations of the simulator will include noise sources whose characteristics fall in-between, such as 1/f noise and non-linearities.

*CIM chip simulation.* We consider a simulated CIM array to be integrated into a simulation of a whole CIM chip. On the level of the chip simulation, we control how input activations and elements of each layer's weight matrix are routed to the CIM array(s). Algorithm 1 explains in detail the simulated process of executing a convolution operation on a CIM chip in the non-shared setting (i.e. we assume access to an arbitrary number of arrays of fixed size). The execution of a fully connected layer is analogous, without the additional step of flattening the kernel and each input patch.

Depending on the size of the convolutional or fully connected layer in relation to the CIM array(s) size, the CIM chip operates differently. When the kernel volume does not exceed the height of the array, the ADC operates as a comparator to directly produce the binary input to the following layer. See Figure 1 for a visualization of the CIM convolution for this case. Alternatively, the matrix-vector operation is split across several CIM arrays and the partial pop-counts are digitized using the ADC to be summed in digital.

Since ADC evaluations require a lot of energy, a guiding design principle for hardware-software co-design lies in designing neural network architectures with small kernel volumes. We explore CIM-friendly architecture design in Section 5.6. Alternatively to a small kernel volume, increasing the array height reduces the number of costly ADC evaluations at the expense of lower SNR and reduced resolution (c.f. Equation (2) and experiments in Section 5.3).

If the number of output channels exceeds the number of columns in a CIM array, the output channels will be mapped to different arrays and the corresponding input patch presented to each of these arrays. Since CIM noises are assumed to be independent across columns, this horizontal splitting of the kernel is not explicitly modeled in the non-shared scenario.

In the shared CIM array scenario, however, Algorithm 1 is modified slightly. Firstly, after the execution of a layer, its kernel needs to be unloaded to free the array for the execution of the subsequent layer. Secondly, horizontally splitting the kernel requires unloading and loading these different kernel parts and computing all output feature maps in multiple steps. In the non-shared case, process variation noises $\alpha$ and $\beta$ induce systematic error only across the entries of a feature map. In the shared case, however, the same column is re-used multiple times also between layers, between different feature maps (horizontal splitting), and across partial convolutions when splitting vertically. In the experimental section, we show that this systematic error for a given CIM chip influences predictive performance significantly.

## 3.2 Hardware Aware Training

The guiding principle for training neural networks for CIM application is to expose, during training, the neural network to the environment it will be exposed to at test time. If a network is trained to perform well across the distribution of all possible chips during training, then it will perform well on a test chip that is drawn from the same distribution as the multitude of simulated CIM chips during training, provided they approximate reality well enough.

---

**Algorithm 1** Binary non-shared CIM propagation; Evaluation

---

**Require:** $x \in \{0, 1\}^{c_{in} \times h \times w}$: Binary input. $w \in \{0, 1\}^{c_{out} \times c_{in} \times k \times k}$: Binary weights. $N, M$: Height, width of CIM array(s). $b$: ADC/DAC bit-width. $\theta \in \mathbb{R}^{c_{out}}$: Thresholds.

$L = k \cdot k \cdot c_{in}$: Flattened input patch size.
$K \leftarrow \lceil L/N \rceil$: Number of required CIM arrays.
Split $w$ into $K$ parts $w_q \in \{0, 1\}^{c_{out} \times c_{in,q} \times k \times k}$.
**for** q in K **do**
    $\alpha'_q \sim \mathcal{N}(0, \sigma^2_\alpha), \beta'_q \sim \mathcal{N}(0, \sigma^2_\beta)$: Process variations.
    $L_q \leftarrow k \cdot k \cdot c_{in,q}$: Volume of partial kernels.
    $w_q \leftarrow reshape(w_q; L_q, c_{out})$
**end for**
**for** input patch $x_p$ **do**
    $x_p \leftarrow reshape(x_p; L)$ Flatten input patch
    **if** $L > N$ **then**
        Split flattened $x_p$ into $K$ parts $x_{p,q} \in \{0, 1\}^{L_q}$
        **for** q in K **do**
            Present $x_{p,q}$ on word lines
            $o_q = PCNT_i^{L_q}(XNOR(x_{p,q,i}, w_{q,i}))$
            $\gamma'_q \sim \mathcal{N}(0, \sigma^2_\gamma)$: I.i.d noise.
            $\tilde{o}_{p,q} = \alpha'_q \cdot o_q + \beta'_q + \gamma'_q$
        **end for**
        $\tilde{o}_p \leftarrow \sum_q^K ADC_b(\tilde{o}_{p,q})$: Digitize and sum in digital.
        $\hat{h}_p \leftarrow \tilde{o}_p > \theta$: Digital threshold comparison.
    **else**
        Present $x_p$ on word lines
        $o_p = PCNT_i^L(XNOR(x_{p,i}, w_i))$
        $\gamma' \sim \mathcal{N}(0, \sigma^2_\gamma)$: I.i.d noise.
        $\tilde{o}_p = \alpha'_0 \cdot o + \beta'_0 + \gamma'$
        $\hat{h}_p \leftarrow \tilde{o}_p > DAC_b(\theta)$: Comparator.
    **end if**
    Store feature map entries $\hat{h}_p$ corresponding to $x_p$.
**end for**
**return** $\hat{h}$

---

This argument also applies to low-bit quantization. Here, we choose PBNN [14] for training binary layers and RQ [12] for multi-bit quantization of a network's first layer. At the core of both methods lies the idea of modeling a network's weights as random variables, whose support lies in the possible values that the quantized weight can take on at test time. Instead of quantizing weights during training directly, a probabilistic treatment circumvents the need for the biased straight-through estimator [2] for gradient computation. Instead, RQ uses the concrete distribution [8, 13] to sample weights while slowly annealing the variance of the distribution during training. PBNN avoids sampling by directly approximating the result of the layer's linear operation with a Gaussian distribution (central limit theorem). For the integration of CIM aware training with quantization aware training, we focus in this work on the binary layers alone. We leave fine-tuning multi-bit layers to future work, which requires defining the partial derivative of each bit of an integer's binary representation with respect to the integer value.

For the training of high-performing binary models for CIM devices, we extend PBNN to be CIM aware. Algorithm 2 describes the CIM training procedure from the view-point of an implementation in a deep learning framework.

During training with PBNN, weights $\hat{w}$ and activations $\hat{x}$ are assumed to lie in $\{-1, +1\}$. As such, we need to map the Gaussian pre-activations $\hat{a} \sim \mathcal{N}(\mu_a, \sigma^2_a)$ that are a result of the CLT approximation, to a pop-count of XNOR operations performed in the $\{0, 1\}$ domain for weights $w$ and activations $x$. At test time, the transformation between domains is described by (3) for a kernel $j$ - corresponding to a column $j$ in a CIM array. At training time, the equivalent Gaussian pre-activation pop-count $a$ is obtained by applying the same transformation to the Gaussian pre-activations $a = \mathcal{N}(\frac{\mu_a}{2} + \frac{N_{in}}{2}, \frac{1}{4}\sigma^2_a)$.

$$PCNT_i^{N_{in}}(xnor(x_i, w_{ij})) = \frac{1}{2}\sum_i^{N_{in}} \hat{x}_i \cdot \hat{w}_{ij} + \frac{N_{in}}{2} \qquad (3)$$

After mapping PBNN computation to the equivalent CIM array computation, we can proceed to integrate the simulation noise model described by Equation (1). During forward propagation, the noise sources $\alpha$, $\beta$ and $\gamma$ are sampled and applied to a feature map of Gaussian pre-activations $a$.

As part of taking a systems view on the simulation, we explicitly model the situation in which the length of the flattened kernel exceeds the number of rows $N$ in a CIM array. In this case, we simulate the ADC and quantize $a$ to $2^b$ evenly spaced values between 0 and $N$. During training, $a$ is a Gaussian random variable, requiring us to either sample from $a$ using the reparameterization trick [9] before rounding or to consider probabilistic alternatives such as RQ. In preliminary experiments for the range of bit-widths we experimented with (c.f. Section 5.5), we observed no advantage from using RQ. Therefore, we use sampling in combination with the straight-through estimator and avoid the computationally expensive probabilistic relaxation of RQ. Upon adding all quantized partial pre-activations, we undo the transformation of (3) and formulate the probability of stochastic binary activations as the difference to the threshold $\theta$. In case of a sufficiently large CIM array, we proceed with the PBNN binarization procedure of computing the probability mass that lies above the threshold $\theta$.

## 3.3 Multi-bit computations

As we show in Section 5.5, applications require the input and first layer's weights to be in multi-bit precision to avoid unacceptable loss of predictive performance. In practice, it will depend on the input size, required bit-widths, and power profiles of the hardware whether or not the first layer should be executed as part of an upstream digital processor or within the CIM chip. A convolution with input and kernel quantized to a uniform grid can be implemented as integer multiplications and additions with proper treatment of grid scales and zero-offsets. See [7] for a discussion of the involved arithmetic. For integer convolution on binary CIM hardware, an integer convolution can be implemented as a weighted sum of binary convolutions as formalized in Equation (4). For the $i^{th}$ scalar entry in $x$, $x_{i,b_x-1}$ corresponds to the value of the last bit in its binary representation consisting of $b_x$ bits. On CIM hardware, each binary operation is executed on a CIM array, digitized using ADCs and

---

**Algorithm 2** Binary non-shared CIM propagation; Training

---

$\hat{x} \in \{-1, +1\}^{c_{in} \times h \times w}$: Binary input. $w \in \mathbb{R}^{c_{out} \times c_{in} \times k \times k}$: Binary weight probabilities. $N, M$: Height, width of array(s). $b$: ADC/DAC bit-width. $f$: Convolutional operation. $L = k \cdot k \cdot c_{in}$: Flattened input patch size. $K \leftarrow \lceil L/N \rceil$: Number of required CIM arrays. $\theta, w = batch\_normalization(x, w)$ **if** $L > N$ **then**

  Split $\hat{x}$ into $K$ parts $x_q \in \{-1, +1\}^{c_{in,q} \times h \times w}$
  Split $\tilde{w}$ into $K$ parts $\tilde{w}_q \in \mathbb{R}^{c_{out} \times c_{in,q} \times k \times k}$
  **for** q in K **do**
    $L_q \leftarrow k \cdot k \cdot c_{in,k}$: Volume of partial kernel.
    $a_q = \mathcal{N}(f(\hat{x}_q, w_q), f(\hat{x}_q^2, (1 - w_q)^2))$: CLT.
    $a'_q \sim a_q$: Sample partial Gaussian pre-activations.
    $o_q = \frac{1}{2} a_q + \frac{L_q}{2}$: Transform to pop-count equivalent.
    $\alpha'_q \sim \mathcal{N}(0, \sigma_\alpha^2), \beta'_q \sim \mathcal{N}(0, \sigma_\beta^2), \gamma'_q \sim \mathcal{N}(0, \sigma_\gamma^2)$
    $\tilde{o}_q = \alpha'_q \cdot o_q + \beta'_q + \gamma'_q$
  **end for**
  $\hat{o} = \sum_q^K ADC_b(\tilde{o}_q)$: Digitize and sum pop-count.
  $\hat{a} = 2 \cdot \hat{o} - L$: Transform to $\{-1, +1\}$ domain.
  $\hat{h} = Bin(\hat{a} - \theta)$: Stochastic binary activations.
**else**
  $\mu_a, \sigma_a^2 \leftarrow f(\hat{x}, w), f(x^2, (1 - w)^2)$: CLT
  $\mu_o, \sigma_o^2 \leftarrow \frac{1}{2} \mu_a + \frac{L}{2}, \frac{1}{4} \sigma_a$: Pop-count equivalent
  $\alpha' \sim \mathcal{N}(0, \sigma_\alpha^2), \beta' \sim \mathcal{N}(0, \sigma_\beta^2), \gamma' \sim \mathcal{N}(0, \sigma_\gamma^2)$
  $\mu_{\tilde{o}}, \sigma_{\tilde{o}}^2 \leftarrow \alpha' \mu_o + \beta' + \gamma', \alpha'^2 \sigma_o$
  $\mu_{\hat{a}}, \sigma_{\hat{a}}^2 \leftarrow 2 \cdot \mu_{\tilde{o}} - L, 4 \cdot \sigma_{\tilde{o}}^2$: To $\{-1, +1\}$ domain.
  $\hat{h} = Bin(1 - \Phi(DAC_b(\theta), \mu_{\hat{a}}, \sigma_{\hat{a}}^2))$: Stoch. activations.
**end if**
**return** $\hat{h}$

---

in-digital scaled and summed for computation of the final result.

$$\sum_i^{N_{in}} x_i \cdot w_i = \sum_i^{N_{in}} (2^{b_x - 1} x_{i, b_x - 1} + \cdots + 2^0 x_{i, 0}) \cdot$$
$$(2^{b_w - 1} w_{i, b_w - 1} + \cdots + 2^0 w_{i, 0}) \quad (4)$$

$$= \sum_{j=0}^{b_x - 1} 2^j \sum_{k=0}^{k} 2^{b_w - 1} \sum_i^{N_i n} x_{i, j} \cdot w_{i, k} \quad (5)$$

## 4 RELATED WORK

Hardware-aware training of machine learning models has been discussed in a variety of works [5, 15, 21, 22]. [5, 19, 22] discuss how to fine-tune a neural network to a specific instance of manufactured hardware. Whereas such a strategy can be expected to result in a very high-performing model, it would incur high overhead during manufacturing. Furthermore, any deployment of a new neural network to that chip after manufacturing would require on-device training [5] including the transfer of the dataset.

This work relates most closely to [21] and [15]. In [21], the authors introduce a stochastic hardware model for a binary CIM array based on MRAM. They study the impact of stochastic hardware on the predictive performance of models trained with or without noise awareness. Crucially, the authors consider arbitrarily sized

arrays, an assumption that does not hold in practice. In this work, we show the importance of extending the CIM array model to a model of a CIM chip, taking array sizes, systematic errors, and ADC quantization noise into account.

The authors of [15] consider independent Gaussian noise added to every linear operation in a recurrent neural network that is motivated by a general "neuromorphic" computing chip. They show how applying noise during training makes the model robust during test time, but make no concrete assumptions about the hardware.

## 5 EXPERIMENTS

We study the CIFAR-10 dataset on a small VGG architecture described in [3]. For an application that is more relevant to deployment on CIM hardware, we include experiments with the speech commands dataset [20] on the *tpool2* architecture [17]. Finally, we also present a new architecture for the speech commands dataset that is optimized for deployment on CIM hardware. VGG results are reported on the validation set. Whenever a VGG model is evaluated in a CIM setting, we report the average and standard deviation across 20 evaluations of the dataset on a given model. For the speech commands dataset, we perform an 80:10:10 split into training, validation, and test sets and report accuracies across 20 evaluations. In this CIM configuration, every data point is evaluated on the simulation of a different CIM chip. Since the exact magnitude of the noise sources cannot be disclosed, we assume $\bar{\sigma}_\alpha$ and $\bar{\sigma}_\beta$ to be the most likely values upon testing of a manufactured chip. Unless otherwise mentioned, $\sigma_\alpha = \hat{\sigma}_\alpha$, $\sigma_\beta = 3 \cdot \hat{\sigma}_\beta$ for accentuated noise effect, $\sigma_\gamma = 0.5 \cdot \hat{\sigma}_\beta$ and each models' first layer is assumed to be computed in digital high-precision. For the shared CIM approach, we assume $M = 128$. For each of the CIM-aware trained models, we fine-tune a pre-trained no-noise binary model. In line with [14] we perform batch norm reestimation before evaluating a model.

To study the properties of CIM hardware for neural networks we consider three scenarios: CIM arrays with the same size as a layer's kernel volume ($N = N_{in}$), CIM arrays with a fixed number of rows (**non-shared**) and a CIM chip with one shared CIM array of fixed size (**shared**). See Section 2 for a motivation for these scenarios.

First, we discuss the ability of CIM-aware training to provide models that are robust to CIM hardware execution. Subsequently, we show how the simulator can be used to understand CIM noise characteristics and drive decision making during hardware design.

### 5.1 CIM-aware training

In Tables 1 and 2 we see how models trained with a specific configuration of the simulator behave in other configurations. The first row shows how a non fine-tuned model suffers greatly from being deployed in a CIM environment. Comparing with the diagonal entries, we see that CIM-aware fine-tuning can recover performance to different degrees. The performance degradation in the $N = N_{in}$ scenario is due to the low signal-to-noise ratio of computations for large $N_{in}$ (c.f. Equation (2)): $N_{in}$ for VGG small, for example, ranges from 1152 in the second convolutional layer to 8192 in the first fully connected layer. For $N = 512$ (and $N = 1024$), the SNR is fixed and comparably high. The SNR for the non-shared scenario is influenced by ADC quantization noise (Figures 3a and 3b) in addition to the SNR from the bit-line resolution. The shared CIM scenario benefits

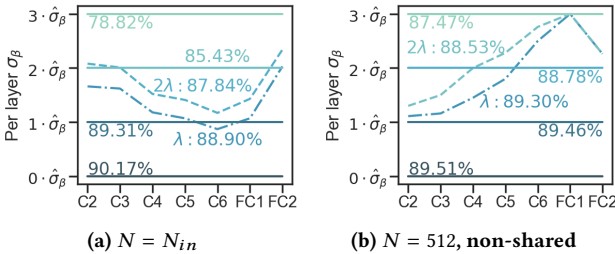

**(a)** $N = N_{in}$

**(b)** $N = 512$, **non-shared**

**Figure 2: CIFAR-10 validation accuracy on VGG small for each noise configuration is written next to each line. Higher accuracy is colored darker. Learned offset results are plotted for two strengths of regularization $\lambda$. The x-axis enumerates the CIM executed layers of the VGG small model.**

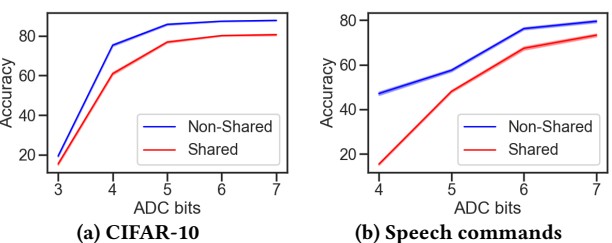

**(a) CIFAR-10**

**(b) Speech commands**

**Figure 3: ADC bit-width variations**

from the increased SNR, however, the systematic error across the network significantly reduces the model's predictive performance.

Even though CIM-aware training can recover a large fraction of the lost accuracy, the difference to the non-shared scenario emphasizes the potential benefits of providing more than one CIM arrays per chip to avoid systematic errors to some degree.

## 5.2 Optimizing for $\sigma_\beta$

Here we show for the VGG small model that a differentiable hardware simulator can be used to drive decision making in the hardware design or manufacturing process. We assume that the magnitude of $\sigma_\beta$ can be influenced by modifying the amount of resources spent (consider offset calibration during manufacturing or enhanced circuit designs). In the following experiments, we make $\sigma_\beta$ a learnable parameter per CIM array (initialized at $\hat{\sigma}_\beta$) and add to the cross-entropy loss function the KL-divergence of $\mathcal{N}(0, \sigma_\beta)$ to the maximum noise distribution $\mathcal{N}(0, 3 \cdot \hat{\sigma}_\beta)$ with regularization strength $\lambda$. The likelihood maximization objective will try to reduce $\sigma_\beta$, whereas the KL-divergence will try to increase it.

In Figure 2a, we see that for the same regularization strength, the $N = N_{in}$ scenario tolerates less noise across the network's middle layers. Where in the $N = 512$ case these layers have a bounded SNR, in the $N = N_{in}$ case, the large number of required rows for those layers makes the model less tolerant to $\sigma_\beta$. The fact that some layers shown in Figure 2b are more tolerant to noise than others might be exploited by spending significant resources only on some CIM arrays on a chip and map the noise-tolerant layers to the less expensively calibrated arrays.

Table 3 shows the impact of different magnitudes of $\hat{\sigma}_\beta$ in the shared scenario. The optimization procedure has to find a trade-off for the noise magnitude between all layers, depending on $\lambda$.

## 5.3 Column length

Increasing the number of rows $N$ in a CIM array reduces the signal-to-noise ratio in exchange for a smaller number $K$ of required ADC evaluations for a given neural network layer with kernel volume $L$ (c.f. Algorithm 1). Table 4 shows the effect on accuracy of doubling the number of rows from 512 to 1024. For the VGG small model, this amounts to a 44% reduction in ADC evaluations. The cost of reducing ADC evaluations is a reduced SNR which in turn causes a non-negligible drop in performance.

## 5.4 ADC bit width

Energy expenditure for ADC evaluations scales with the required bit-width [11]. Determining the minimal required bit-width therefore is an important factor in optimizing the AI performance per Watt of the CIM chip. Figure 3 shows the impact of reducing the ADC bit width on accuracy. A minimum of 6 bits is necessary to avoid a significant drop, which is in line with existing analyses [18].

## 5.5 First layer quantization

As part of taking a holistic systems view on CIM aware neural network training, we address the common assumption in quantized neural network training to keep the first layer (and often also the last layer) of a network in high precision. The required bit-width for the data and the first layer's weights will differ depending on the dataset and model. Here, we investigate the required bit-width for the first layer in VGG small and the *tpool2* architecture, while keeping the rest of the model binary. For CIFAR-10, images are already quantized to 8 bits. Figure 5 shows the impact on validation accuracy of quantizing only the input or only the first layer's weights. Tables 6 show the impact of quantizing both, input and first layer weights for a selection of bit-width combinations. Dependent on accuracy requirements in relation to compute resources, practitioners will have to choose among possible bit-width combinations.

Those results assumed digital computations. Now we evaluate the VGG model on a CIM chip, including the multi-bit first layer. In Section 3.3 we discussed how multi-bit convolutions can be mapped to binary CIM hardware. For all experiments in Table 5, the model has been trained for the respective bit-with of the first layer and the other binary layers, however without CIM-aware training. Since we focus on the first layer's influence, we keep the rest of the model's CIM hyperparameters fixed.

When computing the quantized first layer in digital, the model achieves $89.28 \pm 0.31\%$ accuracy. The second row shows the impact of only the multiplicative noise $\alpha$. Whereas in the binary layers of the VGG small model, the pop count for a CIM array usually falls into the middle of the range between 0 and 512 (assuming approximately half of the feature map entries to be 0), each of the first layer's $6 \times 2$ binary convolutions falls into the range $[0, 27]$. Since $\alpha$ is data-dependent, each of these binary convolution falls into a regime in which the multiplicative noise is larger, given the assumed hardware model. The remaining entries in Table 5 show how adding i.i.d noise $\gamma$ in combination with ADC quantization noise and systematic error $\beta$ influence performance to different extends. In the end, the effects of these noise sources in contrast with the energy expenditure of a digital alternative will depend on the underlying hardware model.

**Table 1: Validation accuracy of binarized VGG small on CIFAR-10. Training scenarios vs. evaluation scenarios.**

|  |  | Evaluated on | | | |
|---|---|---|---|---|---|
|  |  | No Noise | $N = N_{in}$ | $N = 512$, non-shared | $N = 512$, shared |
| **Trained on** | No noise | 90.36% | 74.21 ± 0.87% | 86.32 ± 0.39% | 70.13 ± 0.38% |
|  | $N = N_{in}$ | 90.00% | 78.82 ± 0.73% | 89.57 ± 0.20% | 76.79 ± 0.39% |
|  | $N = 512$, non-shared | 90.42% | 76.98 ± 0.48% | 87.47 ± 0.17% | 71.61 ± 0.29% |
|  | $N = 512$, shared | 89.50% | 79.04 ± 0.48% | 86.07 ± 0.37% | 80.21 ± 0.21% |

**Table 2: Validation accuracy of binarized *tpool2* on the speech commands dataset. Training scenarios vs. evaluation scenarios.**

|  |  | Evaluated on | | | |
|---|---|---|---|---|---|
|  |  | No Noise | $N = N_{in}$ | $N = 1024$, non-shared | $N = 1024$, shared |
| **Trained on** | No noise | 86.33% | 72.10 ± 0.60% | 65.76 ± 0.67% | 43.87 ± 0.85% |
|  | $N = N_{in}$ | 85.09% | 78.75 ± 0.55% | 70.38 ± 0.55% | 55.47 ± 0.68% |
|  | $N = 1024$, non-shared | 83.59% | 76.60 ± 0.42% | 76.21 ± 0.49% | 59.26 ± 0.68% |
|  | $N = 1024$, shared | 79.11% | 73.27 ± 0.55% | 70.21 ± 0.53% | 67.36 ± 0.72% |

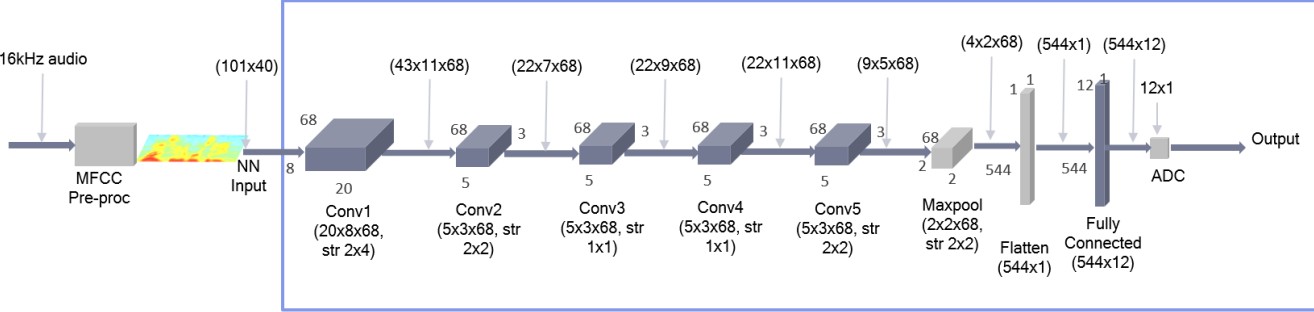

**Figure 4: CIM-NN Architecture. The upper numbers describe feature map dimensions after every layer (with padding). The numbers below characterize the height, width, and the number of channels, as well as the stride of the (convolutional) layers.**

**Table 3: Validation accuracy of small VGG model on the CIFAR-10 dataset in the shared scenario. We compare models trained with different noise magnitudes $\sigma_\beta$, as well as learned noise magnitudes under regularization.**

| $\sigma_\beta$ | Accuracy |
|---|---|
| $0 \cdot \hat{\sigma}_\beta$ | 89.54 ± 0.25% |
| $1 \cdot \hat{\sigma}_\beta$ | 88.28 ± 0.26% |
| $2 \cdot \hat{\sigma}_\beta$ | 85.39 ± 0.32% |
| $3 \cdot \hat{\sigma}_\beta$ | 80.21 ± 0.21% |
| $1 \cdot \lambda : 1.44 \cdot \hat{\sigma}_\beta$ | 87.06 ± 0.33% |
| $2 \cdot \lambda : 1.81 \cdot \hat{\sigma}_\beta$ | 85.74 ± 0.50% |

**Table 4: Validation accuracy of VGG small on CIFAR-10 for different number of rows per CIM array. In brackets: mean accuracy gained through CIM aware training.**

|  | non-shared | shared |
|---|---|---|
| $N = 512$ | 87.47±0.17%(+1.15) | 80.21±0.21%(+10.08) |
| $N = 1024$ | 82.42±0.25%(+2.25) | 73.35±0.42%(+16.59) |

**Table 5: Validation accuracy for CIFAR-10 on VGG small when evaluating a quantized, not CIM-aware trained model in the non-shared scenario. The data and input weights are quantized to 6 and 2 bits. We vary ADC bit-width and $\sigma_\beta$ for the first layer. Other layers are set to $\sigma_\beta = 3 \cdot \hat{\sigma}_\beta$ and $b = 6$.**

| Scenario | Accuracy |
|---|---|
| First layer Digital | 89.28 ± 0.31% |
| $\sigma_\beta = 0, \sigma_\gamma = 0$, no ADC | 57.10 ± 0.61% |
| $\sigma_\beta = 0, \sigma_\gamma = 0.5\hat{\sigma}_\beta$, no ADC | 55.91 ± 0.54% |
| $\sigma_\beta = 0, \sigma_\gamma = 0.5\hat{\sigma}_\beta, b = 8$ | 55.01 ± 0.72% |
| $\sigma_\beta = 0, \sigma_\gamma = 0.5 \cdot \hat{\sigma}_\beta, b = 6$ | 40.04 ± 0.49% |
| $\sigma_\beta = 1 \cdot \hat{\sigma}_\beta, \sigma_\gamma = 0.5 \cdot \hat{\sigma}_\beta$, no ADC | 46.11 ± 0.58% |

**Table 6: Joint weight and input quantization**

| (a) CIFAR-10 | | | (b) Speech commands | | |
|---|---|---|---|---|---|
| $b_x \backslash b_w$ | 2 | 3 | $b_x \backslash b_w$ | 2 | 3 |
| 6 | 90.04% | 90.68% | 6 | 84.41% | 84.02% |
| 7 | 90.04% | 90.40% | 7 | 83.72% | 85.35% |

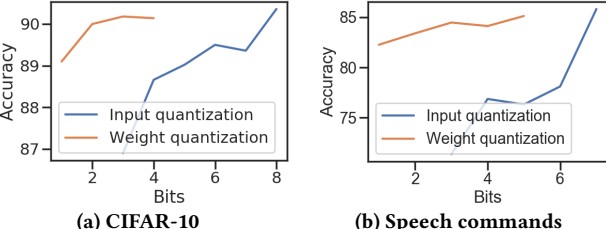

**(a) CIFAR-10**  **(b) Speech commands**

**Figure 5: Separate first layer weight and input quantization**

**Table 7: CIM-NN model. CIM results are fine-tuned from the digital model. We assume $N = 1024$, $M = 128$ in the shared case.**

| Scenario | Accuracy |
|---|---|
| Digital, first layer full precision | 88.50% |
| Digital, first layer $b_x = 4$; $b_w = 2$ | 88.21% |
| Non-shared; first layer full precision | $85.49 \pm 0.33\%$ |
| Shared; first layer full precision | $84.69 \pm 0.45\%$ |

## 5.6 CIM-friendly architecture

Apart from improving the CIM hardware and fine-tuning a given neural network, we can optimize the architecture itself for CIM deployment. Here, we present a CIM-specific network architecture for the speech commands dataset. The major bottleneck we address is the number of required ADC evaluations by reducing the kernel volumes $L$ to be smaller than the number of rows $N$ in a CIM array. Operating ADCs as binary comparators requires less energy and requires only one DAC evaluation for the threshold per input patch (c.f. Algorithm 1). Figure 4 fully specifies the architecture of the new CIM-NN model. At every layer, the kernel volume is less than $N = 1024$ by standardizing the shape of the convolutional filters for all except the first layer. We keep the total number of computations per filter to be less than 1024. This is easy for the first layer, as the input depth is usually 3 or smaller. This is helpful, as we can choose large kernel sizes and strides early in the network to additionally keep the memory requirements for storing binary feature maps less than 3.1KB. Table 7 shows the accuracies of the model under different scenarios. We observe that without the need for ADCs, due to the CIM friendly architecture, the difference in performance drop between the shared and non-shared scenario is reduced. Crucially, the validation accuracy drops only by a small amount on average after fine-tuning (3.01% and 3.81% drop) compared to the original *tpool2* architecture (10.12% and 18.97% drop).

## 6 CONCLUSION

In this work we presented argumentation and empirical justification for a systems view on CIM hardware-aware training of neural networks. We analyzed different aspects of CIM hardware elements and their influence on neural network predictive performance. We showed how a differentiable hardware simulator can be used to inform hardware design and how the co-design of software and hardware for a neural network accelerator allows for informed decision-making along all stages of development. Additionally, we introduced a CIM-optimized neural network architecture for the speech commands dataset. This hardware-aware design of neural

network architectures can be further improved upon by extending recent advances in neural architecture search [4] to the CIM domain. Not only the architecture but hardware properties itself (c.f. Section 5.2) can be made part of the search space. Together, hardware design, neural network architecture design, and hardware-aware training set the cornerstones for optimizing AI performance per Watt.

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
