# OpenReview forum: "Deep Learning for Compute in Memory"
_tinyml.org/tinyML/2021/Research_Symposium — tinyML 2021 Poster_

### Official Review · AnonReviewer3 · 2021-01-27

**Overall Merit Score:** 3

**Brief Summary:**

The paper presents the analysis and exploration of a compute-in-memory (CIM) based hardware performing inference using DNN models. To do so, it also trained DNN models with the consideration of the hardware. The accuracy performances for CIFAR-10 and Google Speech Command are presented, with and without considering hardware noise, and across several design settings (e.g., input quantization)


**Detailed Comments:**

The work is interesting but the treatment of the topic is done is only at a high level. Technical novelty is low. The focus is analysis and evaluation. The spectrum of the evaluation is limited to only two workloads.


**Paper Strengths:**

It is an interesting topic.
Analyzing the input quantization effect is interesting.

**Paper Weaknesses:**

Technical novelty is low as it focuses on high-level modeling. The underlying hardware is treated in an abstract manner.
The hardware is modeled at a high level and thus the paper is mostly focusing on accuracy performance. (no power or throughput)


**Poster (If Paper Is Rejected):**

1: Yes, ok for poster sesion to nurture work

**Reviewer Confidence:**

5: The reviewer is absolutely certain that the evaluation is correct and very familiar with the relevant literature

---

### Official Review · AnonReviewer2 · 2021-01-28

**Overall Merit Score:** 1

**Brief Summary:**

Compute in memory (CIM) has the potential for large energy and latency improvements, but is subject to model accuracy degradation from a variety of noises that impact analog signals. This work aims to explore hardware design choices for preserving model accuracy and robustness. The authors of this work also highlight the importance of considering the systems view for model deployment in CIM chips.

**Detailed Comments:**

There is virtually no novelty in this work and does not include any data pertaining to power or energy (even though it’s a large motivation of the paper). Most of the lessons learned pertaining to CIM array size and AD/DA conversion have already been identified by previous CIM literature (e.g., ISAAC, PRIME, XNOR-Net). This work essentially acts as a summary of various accuracy-power optimization techniques and offers limited new insight into hw-algorithm co-design for CIM deployment.

**Paper Strengths:**

This paper is clear and figures/tables are well made. Experiment methodology is well described. Design decisions are supported by extensive experimentation with good analysis. The identified guiding principles and lessons agree with the literature in the area. The paper makes a good case for optimizing the AI performance-per-watt metric and identifies the systems view as a vital method for achieving it. The paper also extends the CIM array model to consider array sizes, systematic errors, and ADC quantization noise into account. These noise sources are well characterized.

**Paper Weaknesses:**

This paper has several weaknesses that need to be addressed:
-	AI performance-per-Watt was spoken upon heavily in the introduction, but none of the experimentation explicitly shows the model performance-to-power tradeoff. All data showed only consider accuracy, which significantly limits the ability of this paper to claim that its identified design principles are effective at optimizing performance-per-Watt. Please provide power-related data to back up these claims
-	Novelty: This work relies entirely on experimentation to draw lessons/discussions from. It does not propose anything new except for the CIM-friendly architecture; most of the identified design principles are already discussed in previous literature, specifically concerning the ADC component.
-	CIM-optimized NN section is too small, not compared against strong baselines, and not generalized (designed only for the speech command dataset)
-	Too much branching in the text. Most of the paper references Section 5, which is at the end and disrupts the flow of the paper.

**Poster (If Paper Is Rejected):**

1: Yes, ok for poster sesion to nurture work

**Reviewer Confidence:**

4: The reviewer is confident but not absolutely certain that the evaluation is correct

---

### Official Review · AnonReviewer1 · 2021-01-30

**Overall Merit Score:** 2

**Brief Summary:**

This paper investigated the accuracy of CIM hardware considering many design parameters and noise aspects. Evaluations have been carried out for both CIFAR-10 and Speech commands datasets. While not complete, hardware-aware training and architecture designs have been presented.

**Detailed Comments:**

The title of the paper is too high-level. It would be better to describe the specific content and focus of the paper.

The authors mentioned that they consider charge domain computation in SRAM CIM, but there is no detail about the transistor-level schematic. The authors should show the detailed schematic and charge-domain computation, and compare the operation to prior SRAM-based charge domain CIM of [18] and https://ieeexplore.ieee.org/document/9094713 (both works are also XNOR based).

α, β and γ should be defined right after Eq. (1).

For hardware-aware training, the authors should cite and compare to two additional representative works of https://www.nature.com/articles/s41467-020-16108-9 and https://arxiv.org/abs/2001.04974.

For the results of Table 1 and Table 2, the following questions should be answered:
-	What is N_in? The number of rows of CIM array needs to map the entire channels of kernels for each DNN layer? A clear definition is needed.
-	What is the ADC precision for these different CIM schemes with different N?
-	Primarily why does the reported large accuracy degradation occur? In [18] for example, without hardware-aware training, only a very small amount of accuracy difference exists between the algorithm and CIM hardware.

It is not clear how the non-negative activations and weights could effectively map DNNs.

The results of Figure 2 are not explained well.

The abstract mentions energy efficiency, but any quantitative energy analysis is not found throughout the paper.

Why is there an accuracy difference between the non-shared and shared designs? Please explain.


**Paper Strengths:**

Many different design parameters have been comprehensively analyzed and corresponding simulation results have been reported, such as the number of rows in the CIM array, noise characteristics, the first layer precision, etc.

**Paper Weaknesses:**

The authors investigate many different design parameters and noise aspects of CIM hardware design, but it does not seem there is a holistic analysis that encompasses the overall results.
Only a very small VGG model for CIFAR-10 is used throughout the paper as the DNN model, which makes it difficult to evaluate the generality of the reported results.
There is a discrepancy from the actual CIM chips in the literature, which does not necessarily report large accuracy degradation, but this work shows considerable accuracy degradation to the noise in CIM hardware.
The CIM-friendly architecture proposed in Section 5.6 might be causing too much adaptation for general DNN algorithms especially considering many large and wide emerging neural network models, and a large accuracy drop could occur due to this on the algorithm side.


**Poster (If Paper Is Rejected):**

1: Yes, ok for poster sesion to nurture work

**Reviewer Confidence:**

5: The reviewer is absolutely certain that the evaluation is correct and very familiar with the relevant literature

---

### Decision · Program_Chairs · 2021-02-05

**Decision:**

Accept (Poster)

**Comment:**

Based on the reviewer feedback, your paper has been accepted as a poster.

Please read the reviews carefully and make sure the concerns are addressed in your poster submission.

Accepted posters are given a 5-minute slot for an oral presentation on Friday, March 26, 2021, to pitch the main ideas of your work and to stimulate discussions. Detailed instructions will follow soon. All final posters will earn a stamp of acceptance as such: “Published as a poster at TinyML Research Symposium 2021.”